# Dystocia after Unwanted Mating as One of the Risk Factors in Non-Spayed Bitches—A Retrospective Study

**DOI:** 10.3390/ani10091697

**Published:** 2020-09-19

**Authors:** Dejneka Grzegorz Jakub, Ochota Małgorzata, Bielas Wiesław, Niżański Wojciech

**Affiliations:** Department of Reproduction and Clinic of Farm Animals, Faculty of Veterinary Medicine, Wroclaw University of Environmental and Life Sciences, pl. Grunwaldzki 49, 50-366 Wroclaw, Poland; grzegorz.dejneka@upwr.edu.pl (D.G.J.); wieslaw.bielas@upwr.edu.pl (B.W.); wojciech.nizanski@upwr.edu.pl (N.W.)

**Keywords:** dystocia, dog, misalliance, unwanted mating, caesarian section

## Abstract

**Simple Summary:**

This is an article presenting the clinical data on the incidence of difficult labour in dogs being mated without owners’ intentions (accidental mating, unplanned breeding). Despite the widespread availability of spaying and its safety, unplanned and unwanted pregnancies in dogs are still a frequent concern. Unfortunately, in 8.3% (76/914) of cases, they result in difficult labour and deadly complications for the dam and her litter. Better owners’ education about the importance of neutering their pets and understanding of the benefits of early spaying by animal shelter managements and by breeders selling pets might improve the situation in the future. On the other hand, in such cases, it seems to be justified to advise the termination of pregnancy through spaying or the use of abortifacient drugs, despite the ethical concerns, because waiting for natural delivery could cause unnecessary suffering both to the bitch and her unborn puppies.

**Abstract:**

This article presents a retrospective study on dystocia cases in bitches that were unintentionally mated and carried an unwanted pregnancy in the last 39 years. The evaluated medical records include 76 cases of difficult labour, which is 8.3% of 914 dystocia cases recorded during the period. Of these bitches, 38.2% (29/76) were 8 years, and 18.4% (14/76) were younger than 12 months. In 67/76 cases (88.2%), conservative (pharmacological and manual) obstetrical assistance proved to be unsuccessful, and caesarian section (CS) had to be performed, in contrast to the remaining recorded cases of dystocia (in which the pregnancy was intended and expected) when CS was performed significantly less often, in 71.5% (599/838) of cases. In unplanned pregnancies, 46.6% (110/236) of delivered pups were dead compared to only 26.4% (864/3273) dead pups in planned pregnancies. *p* value < 0.05 was considered significant. Despite the widespread availability of the spaying procedure nowadays and its safety, unplanned and unwanted pregnancies in dogs are still a concern in clinical practice. However, throughout the years investigated here, we observed an apparent decrease in the occurrence of dystocia after unintended mating, with much less recorded cases from year 2004 (71 vs. 5). Most probably, this is due to the increasing popularity of surgical castration in both females and males, and rising societal awareness of its importance, giving hope that some improvement in the welfare of dogs has already been achieved.

## 1. Introduction

Intact dogs, as in other animals, if left unattended will breed prolifically to maximise the chances for their genes to survive. Thus, the unplanned breeding of a reproductively intact bitch, during a time when she is fertile, is not an uncommon event. The unexpected pregnancy carries several risk factors for both the dam and her litter. The pregnant female requires not only suitable nutrition, a sensible exercise schedule, appropriate veterinary procedures, and medications, but also proper whelping care and readily available professional help if necessary. Unfortunately, in the case of unexpected pregnancies, these requirements often are not met, or at the very least during only part of the course of pregnancy. The lack of pregnancy awareness usually results in pregnancy monitoring failure and owners being utterly unprepared for the upcoming labour. In such conditions, even a mild delivery complication might become fatal both for the dam and her offspring.

Parturition is the most critical moment during the whole pregnancy, and labour difficulties are frequent problems in canine veterinary practice [1,2,3]. The inability to deliver a fetus through the birth canal is called dystocia. Traditionally, dystocia is classified as being due to maternal or fetal factors, or a combination of both, that hinder the natural progression of fetal expulsion. When dystocia occurs, it should be treated as a life-threatening condition for both the mother and the fetus. It would require prompt veterinary assistance, may cause serious economic losses, and especially in cases of an unwanted pregnancy, it may deeply affect the human–animal bond. The overall occurrence of dystocia in dogs is not high, and in relation to the entire population of this species, it affects only several percentages of births (around 5%) [4], except in some breeds where it may even approach 100% [5,6,7]. Unaware bitch owners might not recognise or may misinterpret the symptoms of upcoming labour, and consequently, they would not establish a collaboration with a veterinary surgeon, resulting in neglect of the pregnant bitch and seeking veterinary assistance much later than necessary, all together contributing to severe medical consequences [8].

While the available literature provides broad information on the incidence of dystocia and its causes in bitches, as well as the individual breed predisposition for impaired delivery, there are no published data that evaluate particular cases of this birth complication as a consequence of pregnancy after unintended mating. The aim of the work was to analyse dystocia cases in bitches that become pregnant against their owners’ will, based on the clinical records collected at the Department of Reproduction, Faculty of Veterinary Medicine in Wroclaw.

## 2. Materials and Methods

The medical records of 914 canine dystocia cases presented at the Department of Reproduction, Faculty of Veterinary Medicine in Wroclaw from 1981 to 2019 were analysed. The analysed cases did not include elective caesarian sections performed before onset of parturition.

These records include 76 cases of delivery complications in bitches where owners declared that pregnancy was a result of unintended and unplanned mating. All of the investigated bitches were privately owned and except for 4/76 cases (5.3%), all were brought in by owners, whereas these 4 were referred by another veterinary practitioner. At the time of admission in each case, there was a strong clinical suspicion of dystocia of different degrees. At that time, all the presented bitches were in the second stage of labour, which had not progressed for at least the last 2 h. Upon presentation, in all cases, a detailed patient’s history including age, breed, parity, litter size, current pregnancy course, labour duration, and its progression was taken. Next, a clinical examination was performed focusing on the general health of the dam and fetuses, the severity of dystocia, and its possible cause. The obstetrical aid necessary to deliver all pups including caesarian section (CS) in each case was recorded and analysed.

In all the cases analysed here, the owners declared clearly that mating was unwanted and accidental and the resultant pregnancy had not been expected or intended.

The data are presented as numbers, percentages, mean value, and standard deviation. Statistical analysis was performed using the chi-square test to compare results between groups of females presented with dystocia after unintended vs. planned matings to check if there was a difference between the dystocia causes. *p* value < 0.05 was considered significant.

Ethical approval was not deemed necessary for this study; the owner signed an informed consent for carrying out all the procedures and treatment necessary in each investigated case.

## 3. Results

The evaluated medical records include 76 cases of difficult labour, which were documented as the result of undesired mating and pregnancy, which is 8.3% of all the dystocia cases (914) recorded during the time period investigated.

From the year 2004, there was a significant decrease in the frequency of the recorded dystocia cases declared to be the result of unintended mating and pregnancy. In years 1981–2003, 71 dystocia cases after unintended matings were noted vs. 679 cases of dystocia after intended matings, whereas in the following years, 2004–2019, only 5 vs. 235 cases were recorded (10.4% vs. 2.1%, *p* < 0.001). Moreover, the average percentage of dystocia in unintended pregnancies in the years 1981–2003 was 10.7%, whereas in years 2004–2019, it was only 1.6% (Figure 1).

The mean age of the investigated bitches was 6.09 (±3.98) years, with the majority of the investigated females (38.2%) being older than 8 years, with one recorded bitch aged 16 years. The second largest group represented dogs that whelped that were younger than 1 year old (18.4%), with the youngest recorded bitch aged 7 months at the time of delivery (Table 1). It was the first delivery for most of the females (58%, 40 bitches), second in 18.8% (13 females), third in 11.6% (8 females), fourth in 7.2% (5 females), fifth in 1.4% (1 female), and sixth in 2.9% (2 females), with the mean parity value at 1.84 ± 1.25 labour. Unfortunately, in 7 of the investigated females, data on parity was not available (*n* = 69).

Moreover, most (52 females, 68.4%) of the investigated dogs were mongrels. The only recorded purebred dogs seen more than once were dachshunds (7 females, 9.2%), miniature pinscher (3 females, 3.9%), and Yorkshire terriers (2 females, 2.6%) (Table 2). The mean litter size was 2.96 ± 1.93, but in over half (54.7%) of the presented cases, pregnancies with only one or two puppies were noticed (Table 3). There was no difference in whelping season and the occurrence of dystocia in the investigated animals: in winter (January–March), there were 22 cases (28.9%) recorded; in spring (April–June), there were 23 (30.3%) cases; in summer (July–September), there were 12 (15.8%) cases; and in autumn (October–December), there were 19 (25%) cases. Unfortunately, in most of the cases, data on the pregnancy course and labour duration was vague or unavailable.

The main reason for dystocia in unintended matings was fetomaternal disproportion (38.2%), followed by single-pup pregnancy (27.3%, in comparison in intending matings, it amounted for only 15.3%, *p* ≤ 0.005) and finally primary uterine inertia (18.4%) classified after Jonhston et al. [8]. Whereas, for intended mating, the main reason for dystocia was the primary uterine inertia (49.1% vs. 18.4%, *p* ≤ 0.0001) (Table 4).

In 67/76 cases (88.2%), conservative (pharmacological and manual) obstetrical aid proved to be unsuccessful and CS had to be performed, in contrast to the remaining recorded cases of dystocia (where the pregnancy was intended and expected) in which CS was performed significantly less often, in only 71.5% of cases (599/838) (Table 5). One of the investigated females had to be excluded from further study because the owner rejected any further assistance and sought a second opinion.

As mentioned above, the analysed cases did not include elective caesarian sections performed before the onset of parturition.

From the remaining 75 bitches, 236 pups were obtained; however, 46.6% were pronounced dead at the moment of birth/delivery, and more than half of the dead ones (61 pups, 25.8%) had obvious post-mortem morphological defects. Whereas, in dystocia cases when mating was planned and intended, only 26.4% of pups were pronounced dead at delivery, and the number of viable pups was significantly higher (73.6% vs. 53.4%, *p* ≤ 0.0001). The average number of newborns for each dystocia parturition in unintended matings was 3.1 vs 3.9 in intending matings, whereas the average number of viable neonates was 1.7 vs. 2.9 (Table 6).

## 4. Discussion

Despite the widespread availability nowadays of spaying and its safety, unplanned and unwanted pregnancies in dogs are still a frequent concern. Unfortunately, most inexperienced owners are not aware that a bitch as young as 6 months of age may already become pregnant. Moreover, during the first weeks, a female dog is unlikely to show any signs of pregnancy, and it is fairly difficult for the owner to realise that the bitch is actually pregnant. Since apparent symptoms such as weight gain or mammary gland development are delayed in time and only visible after fetal development is somewhat advanced, offering pregnancy termination treatment becomes questionable due to ethical concerns. An unintended mating resulting in unwanted pregnancy causes not only ethical or medical implications, but it may also involve severe complications affecting the general welfare of the animal and its litter.

The most dangerous time of the whole pregnancy is birth [3]. Even with experienced and expectant owners, there might be deadly complications if dystocia develops. The risk for dystocia ranges from 5% to 26% [6,9,10] if all breeds are considered. According to most authors [1,11,12,13,14,15], uterine inertia represents the leading cause for birth complications in bitches. In our study, uterine inertia was also the main reason for dystocia in pregnancies after intended mating. However, in unwanted pregnancies, the most common reasons for whelping problems were fetomaternal disparity, followed by single-pup pregnancies and then primary uterine inertia. The frequency of fetal oversize (absolute or relative) attained in the studied group was 17.1%. Most probably, this was due to mismatched male sizes, because the mating was unplanned and unsupervised, or the recorded, rather high incidence of healed pelvic fractures (13 cases or 18.3%) in the bitches studied. A similar percentage of pelvic fractures has been reported in cats [16], which is a common consequence of road traffic accidents involving displacement of the pelvic bones, which can severely obstruct the birth canal [12]. The high number of recorded pelvic trauma in the bitches investigated here raises a suspicion that owners who are unable to responsibly mate their dogs are similarly reckless regarding walking and handling them, thus contributing to traumatic accidents. Furthermore, many of the investigated females were too young (<1 year old, 19.7%) to be bred. Juvenile, developing females should not become pregnant, in order to avoid the increased risk of fetomaternal disproportion [12]. On the other hand, we also noted bitches that were too old for breeding purposes (>8 years, 39.4%). Interestingly, owners of older bitches were truly surprised that their dog became pregnant, assuming that older age alone guarantees contraception. It has been reported that bitches more than 6 years old could have problems conceiving, thus having a lower number of pups, as well as an increased tendency to uterine inertia and prolonged parturition [15]. It is worth noting that in our study, singleton puppy pregnancies contributed to more frequent (27.3%) cases of dystocia compared to reports by other authors (8–19%) [9,11,15,17]. Most commonly, in such pregnancies, the single pup would grow to a larger size, leading to a fetal oversize syndrome and additionally to uterine inertia [7,12,13]. Furthermore, single-pup pregnancies would be more frequently unrecognised by the owners, because the physiological changes are milder and less visible, increasing the chances for unsupervised labour and its associated complications.

The risk for dystocia correlates strongly with the breed, placing the brachycephalic bitches in the most predisposed group [3,4,5,6]. However, in our retrospective study, we identified 13 breeds, and among them only boxers are classified as brachycephalic. Most likely, the large number of specific breeds represented in our study is accidental and was related rather to the popularity of particular pedigree dogs in the given time and area. Nevertheless, it should be noted that over half of the bitches recorded (71.8%) in our study were of mixed breed. It can be assumed that in pedigree bitches (especially valuable ones), the oestrous cycle and the mating time are closely monitored, and the male dogs are chosen very carefully, leaving no room for unplanned reproduction. Such owners are usually more knowledgeable, seek veterinary attention sooner, and most importantly, are better prepared financially to cover the applicable expenses.

According to Traas [18], up to 80% of complications during parturition in bitches could only be resolved by surgical intervention. Many authors confirmed that caesarian section is the method of choice in resolving difficult labour, regardless of previous attempts at conservative treatment [2,9,11,19,20]. In our research, caesarian section had to be performed in most of the recorded cases (88.2%), which exceeds the above cited percentage, as well as the frequency of CS (71.5%) performed in the remaining cases of difficult labour seen in our department during the same time. However, this analysis did not include elective caesarian sections performed before the onset of parturition. The main recorded causes were either fetomaternal disproportion or single-pup pregnancy.

In our investigations, only in one-third of the consulted cases (23/76 females, 32.4%) was the precise time of mating known. This means that in the majority of the females investigated, not only the date of mating, but also the breed and the size of the male dog were unknown. The presented results suggest the need for owners’ education, as the lack of knowledge and sometimes negligence might result in unplanned mating, regardless of the age of the bitch, her hormonal status, and the size or breed of the male dog, which undoubtedly increase the risk of delivery problems. Since most of the bitches claimed to be accidently mated were mongrels, their puppies would not be profit-making, and raising them and the likely accompanying dystocia could be costly, which even further aggravates the clinical situation and worsens the financial constraints. On the other hand, such bitches cannot be left without veterinary care, because birth complications are very likely. It should be noted that almost half (46.6%) of the pups delivered from the bitches investigated were dead, which is much more than described in the available literature, where reported stillbirths were around 4–15% [10,17,19,21,22], with the higher stillbirth percentages occurring the later the obstetrical aid was applied. This corresponds with reports of higher quantities of bacterial growth and mean numbers of bacterial isolates along with prolonged parturition and death of the fetuses [23]. In our study, it may be suspected that owners who did not monitor the heat and mating of their bitches would be similarly careless regarding pregnancy and whelping, and they would not seek professional help in the case of problems with delivery. Overly delayed obstetrical aid in our cases is also consistent with the fact that many puppies (25.8%) had advanced necrotic changes at the time of delivery, whereas in the remaining cases of dystocia when pregnancy was intended, the dead and necrotic fetuses represented only 4.9% of the litters.

## 5. Conclusions

Even though the frequency of 8.3% for difficult labour resulting from unintended mating and unwanted pregnancy appears not to be a large proportion, the risk of deadly complications in such cases is high enough to indicate the necessity of broader consideration and better prevention. Considering these results, it seems to be justified to advise termination of pregnancy through spaying or the use of abortifacient drugs such as aglepristone in these cases, despite the ethical concerns, because waiting for natural delivery could cause unnecessary suffering both to the bitch and her unborn puppies. Our study indicate the urgency for raising public awareness about the importance of neutering pets and for better understanding of the benefits of spaying by owners, animal shelter management, and by breeders selling pets. It is commonly known that after surgical, neutering ovarian and uterine disease cannot occur, nor can disorders related to pregnancy and parturition [8]. Throughout the years investigated, there was an apparent decrease in the occurrence of dystocia after unintended mating, with significant decrease in the recorded cases from the year 2004. Most probably, this is due to the increasing popularity of surgical castration both in females and males, and the rising awareness in society of its importance, giving hope that some improvement has already been achieved.

## Figures and Tables

**Figure 1 animals-10-01697-f001:**
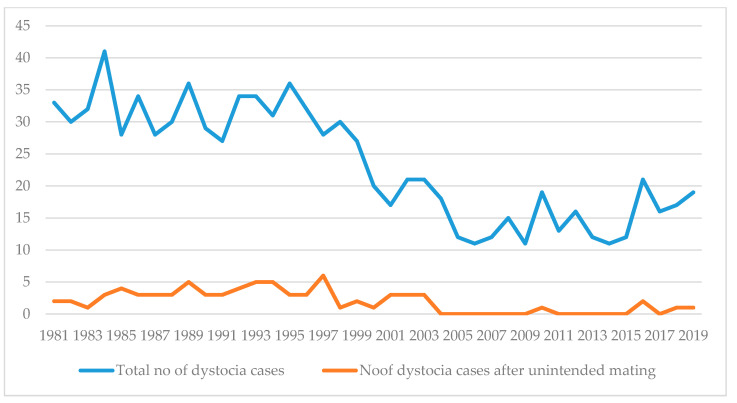
The occurrence of dystocia after unintended mating in the investigated years 1981–2019.

**Table 1 animals-10-01697-t001:** Age distribution in numbers and percentage of bitches presented with dystocia after unwanted mating (*n* = 76).

Age (Years)	No. and % of Bitches
≤1	14 (18.4%)
2	3 (3.9%)
3	7 (9.2%)
4	6 (7.9%)
5	9 (11.8%)
6	3 (3.9%)
7	5 (6.6%)
8	8 (10.5%)
9	4 (5.3%)
10	6 (7.9%)
11	4 (5.3%)
12	1 (1.3%)
>12	6 (7.9%)
**Total**	**76**

**Table 2 animals-10-01697-t002:** Numbers and percentage of particular breeds among examined bitches presented with dystocia after unwanted mating (*n* = 76).

Breed	No. and % of Bitches
Mongrel (Crossbreed)	52 (68.4%)
Dachshund	7 (9.2%)
Miniature Pinscher	3 (3.9%)
Yorkshire terrier	2 (2.6%)
Airedale Terrier	1 (1.3%)
Boxer	1 (1.3%)
Fox terrier	1 (1.3%)
French bulldog	1 (1.3%)
German shepherd	1 (1.3%)
German terrier	1 (1.3%)
German pointer	1 (1.3%)
West Highland White Terrier	1 (1.3%)
Italian Greyhound	1 (1.3%)
Poodle	1 (1.3%)
Siberian Husky	1 (1.3%)
Spaniel	1 (1.3%)
**Total**	**76**

**Table 3 animals-10-01697-t003:** Distribution of litter size of 75 * dystocia cases presented after unintended mating.

No. and % of Bitches	Litter Size
21 (28%)	1
20 (26.7%)	2
9 (12%)	3
8 (10.7%)	4
8 (10.7%)	5
6 (8%)	6
1 (1.3%)	7
1 (1.3%)	8
1 (1.3 %)	9

* one bitch was excluded from further investigation at the owner’s request (see text); no data on newborns available.

**Table 4 animals-10-01697-t004:** Classification of causes (number of cases and percentage) leading to dystocia in 76 bitches pregnant after unintended mating and in 838 bitches mated intentionally (*n* = 914).

Cause	Unintended Mating (*n* = 76)	Intended Mating (*n* = 838)
Uterine inertia		
Primary	14 (18.4%) ^a^	411 (49.1%) ^b^
Secondary	5 (6.6 %)	69 (8.2%)
Fetomaternal disproportion		
Post-traumatic	13 (17.1%) ^a^	0 ^b^
Other	16 (21.1%)	162 (19.3%)
Singleton puppy	21 (27.3%) ^c^	128 (15.3%) ^d^
Prolonged pregnancy	3 (3.9 %)	18 (2.1%)
Vaginal tumour	1 (1.3%)	0
Uterine torsion	1 (1.3%)	5 (0.6%)
Transverse presentation	1 (1.3%)	15 (1.8%)
Monsters	1 (1.3%)	18 (2.1%)
Abnormal posture or position	0	6 (0.7%)
Inguinal hernia of gravid uterus	0	1 (0.1%)

^a,b^—difference statistically significant at *p* ≤ 0.0001; ^c,d^—difference statistically significant at *p* ≤ 0.05.

**Table 5 animals-10-01697-t005:** The total number of dystocia cases recorded from 1981 to 2019 and the number and percentage of caesarean sections performed in bitches after planned (*n* = 838) and unplanned mating (*n* = 76).

	Number of Cases	Number and Percentage of Caesarean Section
Occurrence of dystocia in bitches with planned mating	838	599 (71.5%) ^a^
Occurrence of dystocia in bitches with unintended mating	76	67 (88.2%) ^b^

^a,b^—difference statistically significant at *p* ≤ 0.001.

**Table 6 animals-10-01697-t006:** The general health of newborns obtained from bitches presented with dystocia, in which pregnancy was a result of unwanted mating (pregnancies: *n* = 75 *, neonates: *n* = 236) and in intended matings (pregnancies: *n* = 838, neonates: *n* = 3273).

	Viable Neonates	Dead Neonates	
		Gross Appearance: Normal	Necrotic
Unintended mating	126 (53.4%) ^a^	49 (20.8%)	61 (25.8%) ^a^
Intended mating	2409 (73.6%) ^b^	704 (21.5%)	160 (4.9%) ^b^

* One bitch was excluded from further investigation at the owner’s request (see text), no data on newborns available; ^a,b^—difference statistically significant at *p* ≤ 0.0001.

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
