# Peer review of "Dystocia after Unwanted Mating as One of the Risk Factors in Non-Spayed Bitches—A Retrospective Study"

_animals, 2020, doi:10.3390/ani10091697_

Round 1

Reviewer 1 Report

I have already sent my comments to the authors, who have completed the required changes to the paper.

Author Response

The text have been corrected according to the reviewer suggestions and accepted by the reviewer.

Reviewer 2 Report

In my opinion authors improve the manuscript according to reviewers suggestions. Now the paper is more clear and can be published in Animals

Author Response

(The authors gave the same response as above.)

Reviewer 3 Report

In the attached file you may find some suggestions for a final improvement of M&M and results session. I appreciate a lot the work that authors did during the revision process.

Author Response

Response to Reviewer 3:

Line 11: corrected in text as suggested

Line 25-26: word ‘chi-square’ was removed, P value was added (Line 27)

Line 96: total number of cases was added (Line: 92)

Line 99-100: the following sentences were added: In years 1981 – 2003, 71 dystocia cases after unintended matings were noted vs. 679 cases of dystocia after intended matings, whereas in the following years 2004 – 2019 only 5 vs. 235 cases were recorded (10.4% vs. 2.1%, p<0.001). Moreover, the average percentage of dystocia in unintended pregnancies in years 1981 – 2003 was 10.7%, whereas in years 2004 - 2019 it was only 1.6%

Table 1: as suggested by Reviewer it was changed into graph (Figure 1).

Table 5: was removed and data was added in text as follows: There was no difference in whelping season and the occurrence of dystocia in the investigated animals, in winter (January – March) there were 22 cases (28.9%) recorded, in spring (April – June) 23 (30.3%), in summer (July – September) 12 (15.8%) and in autumn (October – December 19 (25%) cases.

Table 2, 3 and 7 – authors decided not to remove, as it was suggested by other reviewers (now Table 1, 2 and 5).

Line 120: information was added in text as follows: There was no difference in whelping season and the occurrence of dystocia in the investigated animals, in winter (January – March) there were 22 cases (28.9%) recorded, in spring (April – June) 23 (30.3%), in summer (July – September) 12 (15.8%) and in autumn (October – December 19 (25%) cases (Lines 120 – 123).

Line  199 – 202: the ratio was added in text, however table was not removed as it was also suggested by other reviewers (now: Table 5).

Line 120: added  as suggested by the Reviewer

The authors appreciate the time the reviewer spent correcting our text and we are very grateful for all the valuable comments, which improved the paper significantly.

All the recent corrections suggested by the reviewer have been added to the text. However, the authors decided not to remove the Tables 2, 3 and 7 (in the newest version Tables 1, 2 and 5), as such presentation of data was suggested by other reviewers.

This manuscript is a resubmission of an earlier submission. The following is a list of the peer review reports and author responses from that submission.

Round 1

Reviewer 1 Report

neral considerations:  in this paper, the authors introduced a retrospective study on dystocia cases in bitches that were unintentionally mated and carried an unwanted pregnancy. The study includes 76 cases of delivery complications in bitches. The authors justify the study from the limited information available on this observational condition, collecting the medical records of 914 canine dystocia cases presented at the Department of Reproduction, Faculty of Veterinary Medicine in Wroclaw from 1981 to 2019. Although the study is an observational study, it is interesting to reproductive and clinical pathology, and the information provided is enough. Moreover, there are no published data that evaluate particular cases of canine dystocia  as a consequence of  pregnancy after unintended mating.

Title:  The title describes correctly the content of the paper.

Abstract:  The abstract reflects accurately the paper.

Introduction: This section is appropriate.

Methodology: the research design could be improved. This section is too short, especially because some values (e.g. age, breed, parity, current pregnancy course, labour duration and its progression) have very wide ranges. Other important values are not specified, e.g. litter size, breed size, litter number, whelping season).

Under these conditions, it is difficult to distinguish which effects are due to the pregnancy after unintended mating and which ones are due to other interfering conditions.

Why a descriptive statistics and multivariable analysis was not performed?

Results: From an observational point of view, the results are interesting and well-presented, as there is a considerable amount of data and the corresponding tables are clear. From an experimental point of view, however, the paper suffers from the methodological problems listed above.

Discussion: The comments reported in discussion are consolidated with some speculative information in according to data reported in literature. The conclusions are consistent with the results and some personal opinions, related to the need to educate owners about the importance of neutering their pets.

References: the references on main topics are appropriate and well done.

Author Response

Title:  The title describes correctly the content of the paper.

Abstract:  The abstract reflects accurately the paper.

Introduction: This section is appropriate.

Methodology: the research design could be improved. This section is too short, especially because some values (e.g. age, breed, parity, current pregnancy course, labour duration and its progression) have very wide ranges. Other important values are not specified, e.g. litter size, breed size, litter number, whelping season).

The data suggested by the reviewer was added (parity, litter size and whelping season). Unfortunately, clinical data on breed size (especially for mongrels) is was not recorded in each case, so the authors preferred not to add it as it would be only limited number of females. Similarly the data on labour duration and progression was not detailed in the clinical records.

Under these conditions, it is difficult to distinguish which effects are due to the pregnancy after unintended mating and which ones are due to other interfering conditions.

Why a descriptive statistics and multivariable analysis was not performed?

 The descriptive statistics and multivariable analysis were added to the result section wherever possible.

Results: From an observational point of view, the results are interesting and well-presented, as there is a considerable amount of data and the corresponding tables are clear. From an experimental point of view, however, the paper suffers from the methodological problems listed above.

The reviewer suggestions were added into methodology and results section. Authors hope that it would improve the results and its interpretation enough. 

Discussion: The comments reported in discussion are consolidated with some speculative information in according to data reported in literature. The conclusions are consistent with the results and some personal opinions, related to the need to educate owners about the importance of neutering their pets.

 Discussion was changed according to suggestions.

References: the references on main topics are appropriate and well done.

Reviewer 2 Report

I have only some very minor changes. 

Material and Methods

- please include the average age of dogs included into the study

- was US and vaginoscopy examination perform before delivery? 

- how did youo diagnose the uterine torsion?

- was progesteron determined before the decision of sc?

Author Response

I have only some very minor changes. 

Material and Methods

- please include the average age of dogs included into the study

The average age of dogs was added.

- was US and vaginoscopy examination perform before delivery? 

Unfortunately, in most of the investigated cases the pregnancy was unexpected and owners remained unaware of it for quite a long time. The US pregnancy confirmation was performed only in few cases and owners usually did not decide to do any other prenatal diagnostic tests. They tend to leave the animal until natural delivery, seeking veterinary attention only if problems occurred. Since, the female was presented to us in ongoing labour, unless clinically indicated, we did not performed any further diagnostics.

- how did you diagnose the uterine torsion?

The uterine torsion was diagnosed during Caesarean Section. However, during the initial examination the bitch was in discomfort, showed symptoms of acute abdominal pain and tachycardia, which was indicative for such pregnancy complication.

- was progesteron determined before the decision of sc?

No, it was not. The ongoing difficult labour required immediate intervention in most of the cases.

Reviewer 3 Report

The work is aimed to present data about the incidence of difficulties during labor in bitches resulting pregnant without the owner’s intentions or due to mismating.

Despite the interesting topic treated in the paper and the very well written introduction of the work, there are some important points that needs to be clarified before the acceptance of this work.

1.The work is defined in the first lines in three different ways. line 1: case report; title: case study; line 8: review article; line 19: retrospective study. Authors need to clarify this, as this work seems to be much more a retrospective study, it should be written its initial definition. Also the title in the actual version is misleading and  should be changed.

2.Data taken into account and presented in the work are interesting because it is not frequent to have studies on parameters related to dystocia cases and unwanted mating in bitches, presenting comparison with dystocia cases of bitches with a planned pregnancy. However, data are presented in such an unclear way that makes hard to follow the data presentation.

3.Since the simple summary, authors stress the importance and the urgency for the education of owners on the importance of neutering their pets in order to avoid dystocia after unplanned and unwanted mating. However,in order to do that, a correlation between early spaying increase and reduction of certain parameters, like the incidence of dystocia after unwanted mating, should have been done. Without these data, this study does not clearly show anything about the benefit of spaying on reduction of dystocia due to unwanted mating. This is just an author’s opinion that it should remain in the text, maybe increasing its strength with the support of some literature, but it should be stated in a clearer way that it is an opinion and that it should be proved (maybe in a new work?) in a new study designed for that.

Detailed review:

line 14-18: it is just author’s opinion, not something that is clearly showed by data reported in line 11-12.

Line 24-28: your data report of 8.3 % of dystocia cases recorded in unwanted pregnancies is interesting. However, with these premises, if it is performed a comparison with cases of dystocia in planned pregnancies, a much more detailed data collection about cases of dystocia in planned pregnancy will be expected and  should therefore be performed. Also the number of cases and not only the percentages should be reported, at least in the abstract.

Line 29-31: both numbers and percentages should be reported. Also, no mention to the use of some statistical methods used is reported in the abstract. Please specify.

Line 65-67, “This fact prompted us…” a reformulation of this phrase should be done in order to have a more clear description of the aim of the work.

Materials and methods: this section should be reformulated as many informations are lacking about parameters taken into account. It is not clear if all parameters written in lines 77-78 are considered and / or analysed in the statistical analysis.

Line 83-84: description of statistical methods used should be addressed here in more detail: how chi-square was applied? The comparison was performed between which groups? For which parameters? In order to find which statistical differences? 

Results

In general, all data shown in this section should be shown for both groups of planned vs unplanned matings.

A better description of the number of total cases collected, cosidering dystocia cases of both planned or unplanned mating, should be reported.

Line 88-89: The first sentence is redundant and should be either postponed in the discussion section or erased.

92-94: a descriptive statistics of the frequency of cases of dystocia in unplanned matings divided per year should be added in the results, and also associated to cases of dystocia in planned matings. Moreover, it is mentioned ”a significant decrease in frequency of the recorded dystocia cases”: may you provide detailed results of the statistical analysis?

Line 119: numbers instead of number

Line 140-145: in order to have a comparison that gives more strength to data and that helps with further considerations in the discussion session,  it could be interesting if possible to consider the reason for dystocia not only in the unplanned, but also in the planned pregnancies, like you did for the parameter “dead pups” in the following lines and in table 4.

Table 5: can you put in table also the number and percentages of viable-dead neonates obtained in planned pregnancies toghether with the differentiation of normal appearance vs necrotic?

Discussion

Line 209-210: the frequency of fetal oversize  attained in the study group was 18.3%; this percentage was not reported elsewhere in the text, neither in the results nor in the tables, please check it, as this cause difficulties in following the discussion.

Lines 225-226: the data of percentage of singleton as a cause of dystocia is also in this case reported differently to what reported in the table. Please check it.

Line 245-247: also in this case, percentages are different to those reported in the results; is this due to a different count of the cases? Please reformulate this sentence.

Line 249: it is always better to report also the total of cases : 23/76; in any cases these are results that should be reported in the previous section, they should not appear here for the first time.

Line 252-254: These are not results clearly shown by data, but interpretation that authors give to data. Please reformulate.

Line 271: the frequency of difficult labor (7.8%) is different from what reported at the beginning of the work and in the result session. Please, verify your data.

Line 274-276: this sentence is too judgemental and does not derive from any kind of study on the population of owners having experienced dystocia in their dogs. From my point of view it should be erased.

Line 279-282: this study did not clearly showed what is reported in this sentence. Again this is author’s opinion and this should not be confused. All this theory of authors about the responsibility of owners should be transformed to a suggestion of performing future studies about public awareness, needed to clearly show what authors state in the end of the discussion. We have also to not forget that owners education is also Veterinarian’s responsibility to educate owners in order to not to let unplanned mating happen, and this shoud be stressed much more in the text in order to increase this awareness to the public that would read this paper.

Author Response

The work is aimed to present data about the incidence of difficulties during labor in bitches resulting pregnant without the owner’s intentions or due to mismating.

Despite the interesting topic treated in the paper and the very well written introduction of the work, there are some important points that needs to be clarified before the acceptance of this work.

1.The work is defined in the first lines in three different ways. line 1: case report; title: case study; line 8: review article; line 19: retrospective study. Authors need to clarify this, as this work seems to be much more a retrospective study, it should be written its initial definition. Also the title in the actual version is misleading and  should be changed.

Ad.1 The authors fully agree with the reviewer and corrected the misleading definitions into ‘retrospective study’ throughout the text.

2.Data taken into account and presented in the work are interesting because it is not frequent to have studies on parameters related to dystocia cases and unwanted mating in bitches, presenting comparison with dystocia cases of bitches with a planned pregnancy. However, data are presented in such an unclear way that makes hard to follow the data presentation.

3.Since the simple summary, authors stress the importance and the urgency for the education of owners on the importance of neutering their pets in order to avoid dystocia after unplanned and unwanted mating. However, in order to do that, a correlation between early spaying increase and reduction of certain parameters, like the incidence of dystocia after unwanted mating, should have been done. Without these data, this study does not clearly show anything about the benefit of spaying on reduction of dystocia due to unwanted mating. This is just an author’s opinion that it should remain in the text, maybe increasing its strength with the support of some literature, but it should be stated in a clearer way that it is an opinion and that it should be proved (maybe in a new work?) in a new study designed for that.

Detailed review:

line 14-18: it is just author’s opinion, not something that is clearly showed by data reported in line 11-12.

Line 14-18: The sentence was changed into: Better owners’ education about the importance of neutering their pets and understanding of the benefits of early spaying by animal shelter managements and by breeders selling pets might improve the situation in the future.

Line 24-28: your data report of 8.3 % of dystocia cases recorded in unwanted pregnancies is interesting. However, with these premises, if it is performed a comparison with cases of dystocia in planned pregnancies, a much more detailed data collection about cases of dystocia in planned pregnancy will be expected and  should therefore be performed. Also the number of cases and not only the percentages should be reported, at least in the abstract.

Line 24-28: The number of cases was added to the percentage values. Also to clarify the comparison with the dystocia cases in planned pregnancies the following sentence was added to the Materials and method section: The analysed cases did not include elective Caesarean Sections performed before onset of parturition (Line 69 – 70); and a table containing the overview on the number of cases in the investigated years was added to the Results section (Table 1).

Line 29-31: both numbers and percentages should be reported. Also, no mention to the use of some statistical methods used is reported in the abstract. Please specify.

Line 29 – 31: The numerical values and statistical method used were added in the text.

Line 65-67, “This fact prompted us…” a reformulation of this phrase should be done in order to have a more clear description of the aim of the work.

Line 65-67 The sentence has be reformulated into: The aim of the work was to analyze dystocia cases in bitches that become pregnant against their owners will, based on the clinical records collected at the Department of Reproduction, Faculty of Veterinary Medicine in Wroclaw.

Materials and methods: this section should be reformulated as many informations are lacking about parameters taken into account. It is not clear if all parameters written in lines 77-78 are considered and / or analysed in the statistical analysis.

Descriptive statistics was added to the results.

Line 83-84: description of statistical methods used should be addressed here in more detail: how chi-square was applied? The comparison was performed between which groups? For which parameters? In order to find which statistical differences? 

Line 83-84: The following sentence was added in Material and Methods section: Statistical analysis was performed using the chi-square test to compare results between group of females presented with dystocia after unintended vs. planned matings to check if there was a difference between the dystocia causes, its occurrence and neonatal viability in both groups of females (Line 85-88).

Results

In general, all data shown in this section should be shown for both groups of planned vs unplanned matings.

The section was re-written according to reviewer suggestions.

A better description of the number of total cases collected, considering dystocia cases of both planned or unplanned mating, should be reported.

Line 88-89: The first sentence is redundant and should be either postponed in the discussion section or erased.

Line 88-89: this sentence has been removed

92-94: a descriptive statistics of the frequency of cases of dystocia in unplanned matings divided per year should be added in the results, and also associated to cases of dystocia in planned matings. Moreover, it is mentioned ”a significant decrease in frequency of the recorded dystocia cases”: may you provide detailed results of the statistical analysis?

The table containing dystocia cases in the following years has been added to the text (Table 1).

Line 119: numbers instead of number

Line 119: corrected

Line 140-145: in order to have a comparison that gives more strength to data and that helps with further considerations in the discussion session,  it could be interesting if possible to consider the reason for dystocia not only in the unplanned, but also in the planned pregnancies, like you did for the parameter “dead pups” in the following lines and in table 4.

Line 140-145: The data on dystocia reasons in planned pregnancies was  added and the table was reformulated.

Table 5: can you put in table also the number and percentages of viable-dead neonates obtained in planned pregnancies toghether with the differentiation of normal appearance vs necrotic?

Table 5: The data was added and the table was reformulated.

Discussion

Line 209-210: the frequency of fetal oversize  attained in the study group was 18.3%; this percentage was not reported elsewhere in the text, neither in the results nor in the tables, please check it, as this cause difficulties in following the discussion.

Line 209-210: authors are very sorry, it should be 17,1% and was corrected in text

Lines 225-226: the data of percentage of singleton as a cause of dystocia is also in this case reported differently to what reported in the table. Please check it.

Line 225-226: authors are very sorry, it should be 27,3%, correction was made in text.

Line 245-247: also in this case, percentages are different to those reported in the results; is this due to a different count of the cases? Please reformulate this sentence.

Line 245-247: Corrected in text. The authors are very grateful for pointing it out, it was overlooked by the authors, and the numbers referred to the preliminary results.

Line 249: it is always better to report also the total of cases : 23/76; in any cases these are results that should be reported in the previous section, they should not appear here for the first time.

Line 249: Corrected according to reviewer suggestions.

Line 252-254: These are not results clearly shown by data, but interpretation that authors give to data. Please reformulate.

Line 252-254 The sentence: ‘The results presented here clearly show that lack of knowledge, irresponsibility and negligence would result in unplanned mating’ was changed into: ‘The results presented suggest the need for owners education as the lack of knowledge and sometimes negligence might result in unplanned mating’

Line 271: the frequency of difficult labor (7.8%) is different from what reported at the beginning of the work and in the result session. Please, verify your data.

Line 271: Corrected, authors are extremely sorry for this mistake.

Line 274-276: this sentence is too judgemental and does not derive from any kind of study on the population of owners having experienced dystocia in their dogs. From my point of view it should be erased.

Line 274-276: The sentence has been removed.

Line 279-282: this study did not clearly showed what is reported in this sentence. Again this is author’s opinion and this should not be confused. All this theory of authors about the responsibility of owners should be transformed to a suggestion of performing future studies about public awareness, needed to clearly show what authors state in the end of the discussion. We have also to not forget that owners education is also Veterinarian’s responsibility to educate owners in order to not to let unplanned mating happen, and this shoud be stressed much more in the text in order to increase this awareness to the public that would read this paper.

Line 279-282: The sentences were changes

Round 2

Reviewer 3 Report

I have appreciated the improvement made to the manuscript by the authors . However, still some work should be done  in order to get the manuscript published. The main concern is about the poor description on statistical methods used, especially a poor report of the results of the statistical tests used in the M&M section. 

Abstract: even though numbers have been added, the different subdivisions  should always be reported to a total, for example the total number of cases (914, written in M&M) should also be presented here.

Line 20: 76 under brackets is not useful here if you do not indicate also the total number 

Line 21: 8,2% (29/76) ....... 18.4% (14/76)

Line 25: 599 out of the total number.

Line 25-27: "(chi square test)"... this is not the way to report a statistical significance of a statistical analysis

Introduction

Line 55: occurrence of dystocia is not high.... please may you provide a number found during bibliographic research

Materials and Methods

Line 71: I would suggest "these records" instead than "this retrospective study"

Line 73: What 5.3% is for? 4 out of 76?

Line 84-88: statistical analysis has been improved, but still not sufficiently.  how parameters have been considered for the statistical analysis? Why chi-square test has been used? Please provide the setting of significance as well as the results of the test applied to your data.

Lie 111: unfortunately... (n=69). This sentence makes nonsense to me. Please refurmulate.

Line 208: Here you mention a total number=838, while in the M&M there was 914 total cases, why?

Line 251: intended

Line 339:not .....not. Please correct.

Author Response

Abstract: even though numbers have been added, the different subdivisions  should always be reported to a total, for example the total number of cases (914, written in M&M) should also be presented here.

Corrected

Line 20: 76 under brackets is not useful here if you do not indicate also the total number 

The total number (914) was added.

Line 21: 8,2% (29/76) ....... 18.4% (14/76)

Changed according to the reviewers suggestions.

Line 25: 599 out of the total number.

Changed according to the reviewers suggestions.

Line 25-27: "(chi square test)"... this is not the way to report a statistical significance of a statistical analysis

The level of significance was added.

Introduction

Line 55: occurrence of dystocia is not high.... please may you provide a number found during bibliographic research

Corrected as follows: The overall occurrence of dystocia in dogs is not high, and in relation to the entire population of this species, affects only several percentages of births (around 5%) [5], except in some breeds where it may even approach 100% [4, 13].

Materials and Methods

Line 71: I would suggest "these records" instead than "this retrospective study"

Corrected to ‘these records’

Line 73: What 5.3% is for? 4 out of 76?

Yes, corrected.

Line 84-88: statistical analysis has been improved, but still not sufficiently.  how parameters have been considered for the statistical analysis? Why chi-square test has been used? Please provide the setting of significance as well as the results of the test applied to your data.

The level of significance was added to the presented results.

Test chi-square is commonly used in clinical research to compare nonparametric data. It is considered appropriate to compare two or more groups, when the outcome is categorical. The authors wanted to analyse the frequency of incidence of particular factors in the investigated two group of dogs. As we had two nominal variables (group of dogs mated intentionally and non-intentionally), and we wanted to find out whether the observed data in one group differed from the other group we decided the chi-square test wold be most suitable.

Lie 111: unfortunately... (n=69). This sentence makes nonsense to me. Please refurmulate.

Corrected.

Line 208: Here you mention a total number=838, while in the M&M there was 914 total cases, why?

Here the authors wanted to compare the data between group of unwanted (76) and intended (838) matings which in total is 914. We changed the table title to make it more clear, the authors agree with the reviewer that it was misleading.

Line 251: intended

Corrected.

Line 339:not .....not. Please correct.

Corrected.